# The Alcohol Industry and Social Responsibility: Links to FASD

**DOI:** 10.3390/ijerph19137744

**Published:** 2022-06-24

**Authors:** Peter Choate, Dorothy Badry, Kerryn Bagley

**Affiliations:** 1Child Studies and Social Work, Mount Royal University, Calgary, AB T3E 6K6, Canada; 2Faculty of Social Work, University of Calgary, Calgary, AB T2N 1N4, Canada; badry@ucalgary.ca; 3La Trobe Rural Health School, La Trobe University, Bendigo, VIC 3552, Australia; k.bagley@latrobe.edu.au

**Keywords:** Fetal Alcohol Spectrum Disorder, beverage alcohol industry, FASD prevention, alcohol and social responsibility, alcohol marketing, alcohol warning labels

## Abstract

Fetal Alcohol Spectrum Disorder is directly linked to the consumption of alcohol during pregnancy. Prevention programs have been targeted at women of childbearing age and vulnerable populations. The beverage alcohol industry (manufacture, marketing, distribution, and retail) is often seen as playing a role in prevention strategies such as health warning labels. In this paper we explore the nature of the relationship between the industry and prevention programming. We consider the place of alcohol in society; the prevalence, social and economic costs of FASD; the ethical notion of alcohol-related harm and then move onto the question of public health partnerships with the industry including the potential conflicts of interests and ethical challenges in such partnerships.

## 1. Introduction

Alcohol as a consumable product has been around for centuries, although it is not a worldwide pattern. Thus, it is often seen as integrated with human history and evolution, although, in current times, a more balanced view exists where its harms and risks are well documented. In this paper we will look at the difficult relationship between the alcohol industry, FASD, and prevention efforts. The relationship is complicated and, we will suggest, not necessarily complementary. In writing this paper, we were mindful that the role of social policy to prevent FASD may well run counter to the goals of the beverage alcohol industry. Effective prevention will impact levels of consumption in a variety of populations that may be at risk of having a child exposed to alcohol. In addition, care needs to be taken to avoid ways in which corporate liability may be reduced by joint programming, such as was seen in the tobacco industry.

The first part of this paper is a contextual discussion of key issues relating to alcohol consumption and social responsibility that considers (i) the place of alcohol in society, and (ii) the prevalence and social and economic costs of FASD. It then examines (iii) the ethical notion of alcohol-related harm and (iv) the impetus for public health entity partnerships with the alcohol industry. This section concludes with a consideration of the potential challenges and contradictions inherent in alcohol industry efforts towards social responsibility.

The second part of the paper provides a more detailed exploration of the alcohol industry’s approach to social responsibility through two case studies drawn from Canadian, Australian, and New Zealand contexts. The first case study, concerning the éduc’alcool program, Québec, is an example of an education and prevention program funded in partnership with the alcohol industry. The second case study presents an overview of a campaign for mandatory pregnancy health warnings on alcohol products. These two case studies demonstrate some of the contradictions and tensions apparent in alcohol industry efforts towards social responsibility, leading to a discussion and conclusion that outlines questions for future consideration with regards to alcohol industry engagement. The cases used illustrate, within this commentary, areas that social policy makers might wish to include when deciding whether or not partnering with the beverage alcohol industry is a wise thing to do.

## 2. Alcohol Consumption, FASD, and Social Responsibility

i.The place of the alcohol industry

Alcohol is the most widely used psychoactive substance in the world. The World Health Organization WHO [1] reports 3 million deaths annually related to the harmful effects of alcohol which is 5.3% of worldwide deaths. The WHO [1] adds there is a connection between alcohol use and at least 200 disease and injury conditions that can occur throughout the lifespan once consumption begins. In addition, there is a substantial social burden arising from alcohol reflected in economic burdens, early death, social losses, and connections to inter-personal violence. Others estimate the impact is higher in the 15–49 years of age population, representing linkage to 10% of deaths [2].

The worldwide average consumption of alcohol is 6.4 L pure alcohol per person annually [1]. As of 2019, the highest average consumption is in Czechia at 14.26 L/per person pure alcohol while Saudi Arabia, Bangladesh, Kuwait, Somalia, and Mauritania report zero consumption [3]. The place of alcohol within a cultural norm is an important feature in how consumption is sanctioned and the establishment of norms often begins in adolescence [4]. Cultural norms can also inhibit alcohol consumption, for example, in countries where this is discouraged or prohibited for religious reasons, but in most countries beverage alcohol consumption is an accepted cultural norm. The sale and distribution of beverage alcohol is regulated, although the degree of such regulation can vary widely.

COVID-19 may have altered patterns of consumption, although early research suggests a mixed pattern. A Canadian study by Shield et al. describes increased consumption when linked with anxiety, depression, loneliness, or loss of employment. A United Kingdom survey found increases, including in heavy episodic drinking [5]. A USA study also found increases [6], whereas mixed results were seen across several countries although moderate- and lower-income country data was scant [7].

A challenge for discussions, such as in this paper, are key assumptions about the place of the beverage alcohol industry within society and the view that a given society has about its role in food, social life, and society in general. The social influence of the alcohol industry may be perceived as a benign or neutral force (for example, alcohol vendors supporting entertainment events or catering to celebratory occasions where alcohol consumption is normalized), and alcohol industry players have become more actively engaged in messaging around responsible alcohol consumption. However, there are issues that remain to be addressed when it comes to the longer-term and cumulative impact of alcohol on society, including complicated health effects such as FASD and other alcohol-related disorders. We posit that the discussion, relative to FASD, is almost certainly taking place in the context of nations that have developed health care (for diagnostic purposes) and markets (for regulatory purposes). In their review, the WHO, drawing upon the work of Rehm et al., [8] and Blas et al., [9] in WHO Alcohol Fact Sheet [10] built a conceptual causal model of alcohol consumption and health outcomes, where the place of alcohol within society has demonstrable links to causes and effects that intersect with FASD. The model links [10] social and individual vulnerabilities which intersect to create alcohol consumption patterns which then lead to health outcomes. Social factors include such things as access to social determinants of health, the place of alcohol within the context of culture and society, including the way that alcohol is distributed and made available within the society. Individual factors include age, gender identity, and familial context, including the approval or disapproval of alcohol use along with the social and economic position of the family and the person. These link to outcomes such as the level of alcohol consumption, health outcomes and mortality, and other consequences. Thus, using this model, the higher the degree of factors coalescing in favor of higher rates of alcohol use and support of the alcohol industry’s place in society, then higher vulnerabilities for alcohol use in pregnancy can be anticipated [10].

ii.The prevalence and cost of FASD

Just as we have challenges with speaking about alcohol consumption on a global level, so to there is difficulty examining the prevalence of FASD. Accurate figures require good assessment and diagnosis which can then be calculated into frequency data. Researchers conducted a thorough analysis of studies that reported prevalence of the disorder amongst children and youth in the general population and used active case ascertainment or clinic-based methods for determining the presence of the disorder that also specified the diagnostic guidelines or case definitions used to determine the diagnosis [11]. In a systematic review, these researchers identified 24 studies involving 1416 children. Based upon that work, they determined a global frequency of 7.7/1000 population. As Table 1 shows, there can be significant variation in frequency of FASD and alcohol consumption per person. Australia’s numbers appear low, for example, but other sources suggest FASD frequencies of between 2 and 5% of the population [12]. Burns et al. [13] report that FASD is underrecognized in Australia nor have prevalence rates been developed. This helps to illustrate the challenges in understanding true prevalence. Although a new diagnostic protocol has been introduced which may change the patterns around identification and diagnosis [14]. The Aboriginal populations of Australia, along with other groups in more remote communities [15] may also be under-assessed due to poor access to services.

This data does not suggest a direction correlation between alcohol consumed in the general population and rates of FASD. This may well be connected to the degree of accurate reporting as well as the frequency of diagnosis. Australia, as noted, has a lower assessment rate which is partially attributed to the reluctance to diagnose and label a child. Burns et al. [13] note that, in Australia, passive surveillance and under reported case ascertainment create a range of FASD prevalence estimates from 0.01 to 0.68/1000. Reid [16] suggests that Australia lacks accurate or reliable estimates of prevalence. In Canada, a more recent study by Popova et al. [17] suggested the Canadian frequency is estimated at 18.1/1000 population using conservative methods or, using sensitivity analysis, as high as 29.9/1000.

Looking forward, Oh et al. [18] note that frequencies are likely increasing in at least some areas of the world, given that it is now estimated that 10% of women consume alcohol during pregnancy. Meanwhile, Popova et al. [17] add a further layer of complexity to understanding FASD prevalence, noting that people with FASD are overrepresented in the systems and are heavy social and economic burdens in society, including children in care (child protection), correctional judicial systems, special education, as well as in Aboriginal populations. They concluded that the estimated prevalence in these sub-populations was 10–40 times higher than the prevalence in the general population. They also concluded this was likely true on a global basis.

Consistent with the discussion above regarding prevalence, Jacob et al. [19] point out that it is very challenging to know the true costs to society as they are challenging to measure across the multiple venues of interaction with an individual that can include health care, justice, education, child intervention, and social supports along with economic and productivity losses related to caregiving of a person with the disorder. However, even with those limitations, they concluded a range of $22,810 to $24,308 (USD) annually. Thanh and Jonsson [20] estimate that the average person with FASD has a life expectancy of 34 years or a lifetime-average cost of $775,540 and $826,472. Canada estimates 3000 babies are born per year with FASD and the population is presently around 300,000 [21] thus resulting in a cost to Canadians of between $6.8 to $7.2 billion annually. Considering that Canada sells about $22.5 billion alcohol per annum as of 2017 [22] which means that annually, about 32% of that revenue ends up being spent to support a person with FASD. This, of course, does not consider the other health costs to society associated with the variety of outcomes identified by the WHO [1].

These studies reveal two important points. Firstly, that current global understanding of FASD prevalence is insufficient but likely to be underestimated, and therefore cause for conservative consideration in public policy. Secondly, that the harms associated with alcohol consumption and FASD may be amplified for vulnerable populations, possibly resulting in clustering of adverse psychosocial outcomes in already disadvantaged communities and increased reliance on public health and social services. Thus, there is cause to consider the balance of responsibility between industry, individuals, and the state (via policy) in addressing concerns raised by FASD.

iii.The ethical notion of harm

Consumers are perhaps always engaged in balancing responsibilities for risk and benefits when using a product. There is the notion that all products have a potential to harm but that the risk is relatively small, such that the consumer is willing to use it. This is not a universally true statement given that a person taking, for example, methamphetamine through an illicit source, is much more prone to accept that harm is inevitable. Some commercially available products (such as tobacco) have long been known to be harmful to most people when used precisely as indicated (i.e., smoking), yet a portion of the population continues to use them. The Centre for Disease Control (CDC) [23] in the USA, estimate that 14% of adults smoke as of 2019 which is down from 20.9% in 2005. Thus, health risks are just one piece of the risk analysis by consumers.

The consumer of alcohol is likely to accept a level of risk knowing that, at certain levels of consumption, harm is likely. This would be more evident with alcoholism as well as higher levels of binge consumption. The consumer might equally feel that there is a level of consumption below which, harm is unlikely. The appropriate level may be signaled by public policy (e.g., health advice on levels of consumption by age/gender), by cultural norms around drinking, or informed by a person’s own subjective analysis of what seems right to them.

This risk analysis may be influenced by media and advertising, including commercially motivated industry suggestions that there is some level of benefit from moderate consumption. This can include sociability, pleasure, relaxation, and gustatory satiation when such things as food and wine pairings are done. It may also be influenced by research seeming to extol the health benefits of consumption, such as the recent analysis by Yoon et al. [24] indicating a possible reduction of cardiovascular disease when alcohol is consumed in low to moderate levels, even though the majority of literature suggests otherwise. This, of course, causes the beverage alcohol industry to differentiate itself from tobacco, with images that alcohol is not seen as harmful when used appropriately (although just what that is might be difficult to establish worldwide). Indeed, this might well shift the burden onto the consumer to make wise and informed choices and thus, e.g., not drive while impaired, overconsume so as to create immediate and long-term physical health problems, or drink in a way that neglects obligations such as work and family. Purchasing a car could be argued to come with similar obligations.

However, a difference with alcohol is that it is inherently linked to harm [25]. As Room and Cisneros Örnberg [25] note, alcohol has consistently been shown to be in the top ten risk factors for disability and death. FASD is a disability that is directly linked to alcohol consumption during pregnancy. The use of alcohol during pregnancy presents a problematic paradigm as it is difficult to regulate, primarily due to principles such as autonomy, freedom, and human rights. From a social policy perspective, this then leads to balancing harm with regulation around access and distribution of alcoholic beverages, relying upon education about harms and the goal of personal responsibility for usage. This approach was part of the UK strategy to manage social and health issues related to alcohol, although its efficacy is unclear [26]. Bonner and Gilmore [26] support an approach that seems to blend the public education, responsible messaging from the alcohol industry, government regulation, and personal responsibility. They point to other attempts at blended approaches such as the European Alcohol Health Forum [27]. Such approaches mobilize the concept of harm as a lever to involve the alcohol industry in efforts to reduce the harm from the use of its product. When a product such as alcohol causes harm that impacts large swathes of the population it is critical to consider the accountability of the industry for adverse health outcomes. De Lacy-Vawdon and Livingstone [28] identify the health burden of non-communicable diseases (NCDs) that are attributed to lifestyle and include the use of products such as alcohol and tobacco. These products are included in the Commercial Determinants of Health (CDoH), being those substances which are profit driven and affect health [28]. The mammoth size and scope of the alcohol industry has the capacity to be visible in every corner of society, and it is suggested that the producers hold major control in the process of marketing [29]. We suggest the opposite must also be true, in varying degrees, with alcohol contributing a negative input to the health equation. This begs the question—is the alcohol industry culpable in relation to FASD?

iv.Alcohol Industry partnerships

Research on the efforts of the tobacco industry to withhold data is well developed based upon massive disclosure of documents through litigation, particularly in the United States [30]. That data showed that the tobacco industry sought to manipulate research, policy and understanding of the harms directly related to consumption of the product [31]. Whether that is true in the alcohol industry is unknown as the same level of disclosure has not occurred. The lead author was a member of the beverage alcohol industry in the early 1980’s through to 1994. During the latter years of that work, was privy to the spirits industry work in social policy areas. The work was focused upon three main themes at the time: (1) the public’s understanding of the alcohol content of a standard drink of spirits, wine, and beer; (2) how early-aged legal drinkers perceived over consumption; and (3) how governments viewed regulating alcohol. The lead author of this paper, had, prior to joining the industry, been in the regulatory side of the retail distribution of alcohol. In that role, insight into government policy was the attempt to balance responsible distribution along with responsible consumption environments while also promoting access to the retail and drinks market. Underlying much of the thinking was the notion that regulation would not solve problematic drinking but could influence it. The consumer was viewed as the ultimate regulator of their own use. In Box 1, an example is given about how social policy can be used to influence access and consumption of alcohol in highly problematic environments, although it should be noted that it cannot eliminate such use [32]. This example shows the nature of influence, not control, and indicates that approaches such as this might not be effective.

Box 1Liquor retailing, social policy, and substance abuse—a case study.The Downtown Eastside (DTES) of Vancouver and Liquor Retailing—A case study drawn from Bailey [32]The DTES has long been a geographical focal point for high-risk behavior including alcohol and drug use. The ‘epicenter’ of the area is the corner of two streets, Main and Hastings. It was on that corner that the British Columbia government operated a liquor store. In the early 1980’s, pressure developed to close that store as it was seen as central to the alcohol problem in the area. Indeed, as has been documented in the work of Dr. Gabor Mate, the population living there is highly traumatized and such trauma is linked to the elevated prevalence of FASD in the population—as adults living with the disorder and in pregnancies.The Mayor and city council in Vancouver were under pressure to seek closure of this store as a way to mitigate the social issues in the area. As Bailey (2021) documents, it was a simplistic approach that failed to consider the highly complex nature of the problems in the DTES. The closing, which did occur, was a ‘supply side’ solution that is reminiscent of prohibition. Cutting access to alcohol would serve to create improvement. The closure of this store would lead to policy review internal to the British Columbia Liquor Distribution Branch (the retailer) regarding how and where retail operations were set up.The DTES retail decision did not improve the outcomes in alcohol consumption in the area, which remains a ‘capital’ for substance abuse in Canada. As Bailey (2021) notes, “Internal documents from a Mayor’s Task Force that was convened in response to the closure indicate that the City of Vancouver acknowledged the futility of this effort at the time. Task force subcommittee members soon admitted that the closure of the liquor store at Main and Hastings was likely to blame for a noted increase in non-beverage alcohol use.” This included the highly harmful “Chinese cooking wine” which had very high salt content along with other products such as Listerine ©.This case study helps to illustrate that, even when working with the regulated alcohol retailing sector, the problems which underlie FASD in trauma-based populations are not easily addressed through supply side approaches. This area of Vancouver has also shown that, when thinking about this population base, alcohol use is typically mixed with other substance use and social conditions such as poverty, homelessness, and violence. This case also illustrates that harm reduction strategies have limited value in high risk, high consumption populations

When thinking of partnering with the beverage alcohol industry in FASD prevention, it is worth recalling that the ultimate responsibility of a corporation is the creation of profits for the owners and shareholders [33]. The history of tobacco shows that can be challenged in terms of holding a corporation or industry accountable for the harm they create (see for example the Canadian Supreme Court Decision *R. v Imperial Tobacco Canada Ltd.* [34]). We have also seen this with the opioid crisis with Purdue Pharmaceuticals pleading guilty related to the marketing of OxyContin © which downplayed, obfuscated, or failed to disclose the addictive nature of the drug [35]. By engaging in social partnerships with health promotion programs regarding FASD, the industry is doing the opposite by stating that there are potential harms to consuming alcohol in pregnancy. On the one side this is positive public education, while on the other it may be protective of the industry who can say, ‘We told you not to consume in pregnancy. We are not responsible for the decision you then made’. Thus, partnerships with the alcohol industry can also offer it liability mitigation. Another factor to consider when connecting prevention messaging with the beverage alcohol industry, is that the public may be less receptive or skeptical of the value of the message when the industry is seen as offering corporate social responsibility communications.

What is clear is that there is potential for conflicts of interest to exist between the beverage alcohol industry’s efforts to mitigate harm and extoll corporate responsibility on the one hand, and the industry’s ability to extract value and increased market share on the other. However, like tobacco and other products associated with ill-health effects, this corporate responsibility coexists with individual and social factors that can confound and obscure what the industry should be doing or not doing to truly contribute to the prevention of harm arising from alcohol. This is even more the case where FASD is concerned, as the ill-effects are deferred and long-term, and not immediately observable or actionable.

Up to this point in this paper, we have focused on positioning the discussion as part of a larger conversation on the linkages between social policy objectives (in this case prevention of FASD) and the possible place an industry such as the beverage alcohol industry might have in achieving that. We argue that there is a built-in conflict of interest as discussed above. Can it be managed? It is noted that global public health policy in relation to alcohol marketing is severely limited [36] and the WHO Working document for development of an action plan to strengthen implementation of the Global Strategy to Reduce the Harmful Use of Alcohol [37] calls for intersectoral collaboration to reduce alcohol related harms. One key policy option includes “improving capacity for prevention of, identification of, and interventions for individuals and families living with Fetal Alcohol Spectrum Disorder” [37], p. 12. This report also highlights the need for interventions for reduction of alcohol marketing to adolescents and to reduce alcohol advertising exposures, which is reported to be a significant concern in alcohol consumption by youth [36].

To begin with, the separation of industry lobbying from prevention is a core principle. Lobbying is designed to achieve profit goals by limiting the degree of regulation, maximizing access to the consumer both in terms of retailing and advertising, as well as a variety of other activities that limit such things as corporate concentration. Prevention efforts start with a recognition of the harms that a product presents and seeks ways to eliminate or minimize those harms. The industry might well argue that prevention is in their interests as they wish to promote responsible use of their products. To take that position enhances the industry’s lobbying as it presents them as ethical manufacturers who can be trusted to behave so in the marketplace. In the lead author’s direct experience, such positioning was indeed the case. That these lobbying and prevention conversations are mixed is well documented by Lyness and McCambridge [33]. These same authors also illustrate that, were it not for money from the industry, prevention programming, including related research, would be harder to muster. In thinking about the ways in which the industry approaches prevention conversations, it is wise to contemplate the seven key messages of the alcohol industry as seen in Box 2.

Box 2Seven key themes of the alcohol industry regarding social policy agendas [38].
Consuming alcohol is normal, social, and enjoyed as part of a balanced lifestyleAlcohol problems are caused by a small group of people who have other problems and therefore cannot handle alcoholResponsible alcohol consumption is part of a healthy lifestyleAlcohol advertising does not lead to an increase in alcohol consumptionInformation and education about responsible use is the best method to protect society from alcohol problemsAlcohol-free and low-alcohol beverages can play a role in reducing harmful drinkingAlcohol problems can only be solved when all parties work together


That is not an argument for lack of engagement with the industry but rather one where the transparency of agendas matters. Were it not for problems with a negative image and liability, would the industry participate in prevention activities? Is industry-derived support tainted by hidden agendas, or problems of perception? The two case studies to follow will explore these questions in more detail.

## 3. Case Studies of Industry Engagement with Social Responsibility

Case Study 1: Éduc’alcool, Québec [39]

Éduc’alcool is a program operating in the Canadian province of Québec. It has a broad mandate to speak about the implications of alcohol consumption. They describe their purpose as follows:

Éduc’alcool is a private, independent, not-for-profit organization. Its members are Para public institutions, alcoholic beverage industry associations and individuals from various milieus (education, journalism, public health, business community), who implement prevention, education and information programs designed to help young people and adults make enlightened, responsible decisions about drinking and the circumstances in which they drink. Éduc’alcool’s commitment is accurately reflected in its slogan: La modération a bien meilleur goût/Moderation is always in good taste” [39].

Their funding is described as follows:

“Éduc’alcool is funded by a levy taken by the SAQ on the sales of its institutional member’s products, which allows them to fulfill their legal obligations under articles 19, 20 and 23 of the Regulation respecting promotion, advertising and educational programs relating to alcoholic beverages and to obtain their certificate of compliance, issued annually by the Régie des alcools, des courses et des jeux. Institutional members who do not sell their products through the SAQ pay their contribution directly to Éduc’alcool” [39]

Their objectives are listed as:○To educate the public in general and young people in particular with regard to drinking.○To provide information on the psychological and physiological effects of alcohol.○To prevent and denounce alcohol abuse and its consequences.○To promote moderation in drinking.○To debunk myths about alcohol and drinking.○To intervene in order to influence drinking contexts.○To conduct and support social and scientific research.○To examine the historical and cultural context of drinking.○To promote the culture of taste as opposed to the culture of drunkenness.

The website lists an independent scientific advisory committee which they describe as, “The Scientific Advisory Council is a consulting body comprised of academic, research and medical experts. Its goal is to advise and support Éduc’alcool in its activities” [34].

Relative to FASD, the website offers no specific mention of the disorder but does provide guidance on alcohol use in pregnancy, *Alcohol and Health: Pregnancy and Drinking: Your Questions Answered* [39]. This information, they state, follows the most recent guidelines from the Society of Obstetricians and Gynecologists of Canada. It serves as an example of using current health and science guidelines to inform about the risks associated with alcohol use and pregnancy.

The membership of éduc’alcool includes representatives from the beverage alcohol community who thus have a direct say on the activities being undertaken. Members agree to promotion of moderation in the consumption of alcohol.

Case Study 2: The Australian Warning Label Debate of 2020

A major approach to messaging regarding alcohol-related harm and FASD has been to place warning labels on beverage alcohol containers and to post warning messages in drinking establishments. This is the notion that the message can be directly linked to those who are most likely to engage in drinking behaviors and may be able to influence the decision around drinking during pregnancy. There is substantial debate about the efficacy of these messages, although they can be framed as doing no harm, acting in a way to educate, and planting the awareness for future considerations.

Warning labels on commercial alcohol bottles have been an espoused as part of FASD prevention strategies for a number of years. Alcohol industry players have participated in this process, but their participation has been self-guided, not necessarily informed by research. This has resulted in inconsistent labelling in the past, with some parts of the industry embracing the responsibility of providing warning labels, and other parts ignoring it. In 2011, the Australian Foundation for Alcohol Research and Education (FARE) commissioned research into the attitudes and perceptions of Australians about alcohol Health Warning Labels. The resulting report [40] demonstrated strong community support for warning labels, with 72% of respondents indicating support for labels to be subject to government regulation (as opposed to 12% supporting the status quo of industry self-regulation). Specifically, the proposed label “drinking any alcohol can harm your unborn baby” was seen by 86% of respondents as important for raising awareness. This report informed a policy position adopted by FARE, which they and others subsequently used when lobbying government and other stakeholders for better labelling, though to little immediate effect.

A new opportunity for concerted action emerged in 2020 with the announcement of a review of the Food Standards Australia New Zealand Act 1991. Warning labels for alcohol fell under the potential jurisdiction of this Act, and the review was a timely opportunity for community stakeholders to advocate for making labels mandatory. Alongside other health groups and community organizations, FARE [40] advocated for mandatory warning labels, as well as asking the review committee to be mindful of industry self-interest when reviewing competing industry submissions. Meanwhile, alcohol industry submissions to the review sought to downplay or negate a need for the introduction of mandatory warning labels. This debate spilled over into the public arena, with open statements (e.g., https://fare.org.au/wp-content/uploads/Open-Statement-from-Public-Health-FINAL.pdf, accessed on 10 May 2022) and media interest in the growing debate over labelling.

In early 2020, the Independent Regulator Food Standards Australia New Zealand (FSANZ) produced an evidence-based proposal for mandatory warning labels that was submitted to the review. Key features of the proposed label included specific warning text and the use of contrasting colors (red, white, and black). At the March 2020 Forum on Food Regulation, however, Food Forum ministers asked that the color coding and specified text be removed, in line with alcohol industry desires. Meanwhile, alcohol industry lobby groups launched a public campaign against the proposed label (called ‘Not This Label’). They argued that they were not against the need for warning labels per se but concerned about the cost to industry of multi-color printing, arguing that this would hurt small producers in particular. This line of argument held sway with some politicians who likewise advocated for labels to be less proscriptive, even though the FSANZ template reflected evidence-based best practice (see details at https://www.foodstandards.gov.au/code/proposals/Pages/P1050Pregnancywarninglabelsonalcoholicbeverages.aspx, accessed on 15 May 2022).

Ultimately, the Ministerial Forum on Food Regulation that was responsible for the review of the Act accepted the FSANZ label, endorsing the advocacy of community and public health groups. They gave industry a generous 3-year window in which to implement the new mandatory labels on their products but did not accept the industry lobbying that sought to use concern for the financial impact on small producers as a pretext for de-powering the public health impact of the proposed label.

## 4. Discussion

The case studies outlined above highlight some of the tensions inherent in the involvement of the alcohol industry in prevention efforts. They illustrate the tensions that exist in partnering with an industry whose needs are contrary to goals that seek to reduce the very use of their product in a variety of circumstances including when pregnancy is contemplated or has occurred. The case studies also show how the key messages are managed (See Box 2). Good prevention efforts may also seek to have those around pregnant women to also not use alcohol as a way to support the mother’s choice. The two case studies looked at together illustrate the conflict inherent in corporate social responsibility agendas, such as previously seen with the tobacco industry. Recall that in 1970 Milton Friedman made the case that the social responsibility of a corporation is to maximize profits [41]. Gilmore, Savell, and Colin [42] make it quite clear that joint efforts with industry will not be done in a way that puts profits at risk. Madden and McCambridge [43] report that the industry seeks to be seen as productive partners in alcohol policy, key players, and socially responsible (see also [44]).

Hawkins, Durrance-Bagal, and Walls [45] argue that the alcohol industry agenda is to not harm their place in the market and to enjoin with policy makers to ensure that does not happen. Further, it adds a favorable image with the public and allows for industry influence in decisions. This is seen in case 2.

The organization described in Case Study 1 has been subject to criticism, due its close relationship with the beverage alcohol industry, from manufacturing, marketing on through to retailing [46]. Éduc’alcool countered that in a letter to the journal publishing Petticrew et al. [46], stating Éduc’alcool is an alcohol education and prevention organization that bases its information on rigorous scientific data and we intend to continue to inform Quebecers according to the best practices [47]. Yet, Éduc’alcool is a clear example of how the industry seeks to ensure that it is at the table to frame and influence how public health messaging about alcohol is managed. Dwyer et al., [44] have clearly shown how the alcohol industry seeks to accomplish this, such as by aiming towards harm reduction as opposed to reduction of usage. As noted above, the industry seeks to maximize the presence of studies that suggest there may be some health benefits to moderate consumption of alcohol even though that research may be inconclusive.

A major goal in the field of FASD is the prevention of the disorder and the harms related to it, which occur over lifetimes. In general, what we know about prevention of general harms from alcohol are related to disease and social harm agendas. For example, alcohol consumption is linked to cardiovascular disease, cancer, metabolic disorders, and liver disease [1,2]. As pointed out by research, messaging here can be difficult for the consumer who must make decisions now for risks that may not materialize for years [46]. Risk and benefit analysis for the consumer is harder when consequences are difficult to connect to the immediate choice.

It is also linked to social harms such as impaired driving, escalation of domestic violence patterns, and family breakdowns. As discussed above, there is the argument that only a portion of the population may be so affected, with another portion of the population using alcohol in a moderate fashion not deemed particularly harmful. This has meant that broad statements of harm are harder to sustain with the general population.

As the above shows, the position of the alcohol industry in harm prevention and that of the academic community is a difficult one where common ground may hard to find. It may seem that reducing the damage done by a product would offer that space, but then both parties would need to agree on what the harm actually is and whether it is caused by the product, social conditions, or combinations. Plus, there would have to be agreement on who might then be responsible for achieving change. This might well be the crux of problem [46,47,48]. The case studies illustrate this tension along with the reality that prevention is not a single step.

This is seen with the issues presented in Case Study 2. A feature of label warnings is that they refer to a risk that is very immediate if there is a pregnancy. However, the risks can also exist in two very different ways. A feature of label warnings is that they refer to a risk that is very immediate if there is a pregnancy, yet women may not consider possible pregnancy when making drinking choices. Some women receive messaging that a moderate or low amount of alcohol in pregnancy is acceptable. The causal pathways of FASD are multiple and complicated [49]. Recent research suggests that even low levels of alcohol exposure in pregnancy pose a risk [50].

There is yet another area where messaging as presently constructed is less likely to be successful. This is with the highly traumatized, heavily consuming populations that are managing many aspects of the trauma legacy through the consumption of substances such as alcohol [51,52]. Here, the risk benefit contemplation is even more challenging. To not drink during pregnancy or when anticipating pregnancy is to take away a highly effective, if also highly harmful, coping mechanism. Alcohol is often used to self-medicate against the harms of trauma. We would argue that real prevention efforts are far more challenging. The health sector would see harm reduction as more realistic with the trauma populations, which is an impossible territory for the alcohol industry to be seen as credible in.

Although we struggle with the clear problem of stigma arising from the “FASD is a 100% preventable disorder” as a prevention message, it strongly links the notion of harm in a more universal way as opposed to fragmenting the discussion that the risks are unique to a specific population. Rather, messaging campaigns such as this often infer, if not directly state, that the risk exists for all pregnant women. The problem with this message is that it implicates women as solely responsible for alcohol use during pregnancy without contextualizing the circumstances under which this occurs.

In an editorial it is argued that the alcohol industry raised specious arguments against the use of warning labels on alcohol use in pregnancy in Australia Smith et al., [53]. They noted that the industry in Australia has stated that there are costs to the implementation of labels and that the industry omits and misrepresent harms arising from their products. It is suggested in further analysis of the corporate social responsibility message of industry based and funded organizations is done in a way that impacts the quality and tone of the messaging [54]. They support the notion of minimizing and distorting harms while enhancing the positive messages. The basis is the power of marketing tools that nudge readers in certain directions. As noted earlier, an industry supported website may indeed have data in harms, such as FASD, available, but make access more challenging to find or push the reader through levels of information that may soften the harm messaging [54].

Petticrew et. al. [55] cite several tools that can be used to accomplish this, as noted in Figure 1. These tools can also be used by the prevention community although they act as pathways to distort the consumer’s understanding of risks that can influence drinking decisions. These authors also note the power of ‘sludges’ which make getting at information more challenging, creating friction between the user and the information sought such that there is a discouraging element at play.

It is suggested that the alcohol industry is complicit in minimizing, obfuscating, and misinforming the public about the risks for women associated with alcohol and pregnancy [54]. They suggest that there are economic risks to the industry that inform how information is presented. We wonder how the industry is going to respond to the emerging data which suggests that the drinking patterns of fathers also has an impact on FASD risks. Suggesting that the male drinking patterns also need to be addressed in relation to pregnancy is vital to successfully expanding the conversation on prevention. This would serve to move the issue from a “women’s” issue to one that involves men and women [56] This expands the economic implications for the industry.

We then come to the question of whether alcohol warning labels are effective. This of course depends firstly upon what they are meant to do—are they to educate? Impact consumption behavior? Cease consumption for anyone contemplating pregnancy? Influence both the possible pregnant women, her partner and those around her? It likely depends upon who you ask. In our view, they should seek to influence all four goals but are unlikely to accomplish any of them on their own. There is a further complication which is to whom are the messages targeted? Aiming towards younger consumers may require specific approaches that would appeal to them [57,58]. Targeting to other populations may also require specific focused efforts which alcohol warning labels may not be sophisticated enough to manage. In our view, this is a specious argument as it assumes that there is but a single messaging tool available.

A particularly interesting experiment is reported which took place in Whitehorse, Yukon Territory and a comparison control site in Yellowknife, North West Territories in Northern Canada [59,60]. FASD was not a target of the experiment but cancer warning, educating on a standard-sized drink and offering information on recommended maximum consumption for men and women in a day and a week of consumption. The alcohol trade associations successfully lobbied to have the cancer warning labels removed from the experiment. The authors note, “The industry’s opposition is understandable: They fear that strong warning labels will shrink their market and erode profits ”…”. However, it quickly became clear that the cancer warning label, in particular, was eliciting the strong response, which is consistent with broader industry positions on this type of health messaging” [48], p. 10. These authors note that the threat of litigation is a tactic that had been used by the tobacco industry.

Another side of the health warning labels is the consumers’ right to know about the potential harms of a product [60], particularly given suggestions that the consumer does benefit from the knowledge [61].

There is a further complication with the alcohol industry and its connection to prevention messaging which has the goal of reducing, and in the case of pregnancy, eliminating usage. Research suggests that the alcohol industry significantly benefits from harmful patterns of consumption [62]. Research also shows that the industry goals with advertising and marketing are not just to direct consumers to specific brands but to also increase consumption overall [63]. Returning to the lead author’s direct experience with the industry, advertising and marketing had three specific objectives: 1. To increase market share; 2. To increase brand loyalty and 3. To increase consumer desire to consume alcohol. As noted, the first two objectives are the typical conversation of the industry. The third is an obvious desire by profit making organizations. Though logically, it is not an obvious goal to accept that a better market share of a shrinking industry is a good outcome over the long term. Rather, the profit profile has better potential where it is possible to increase consumption and market share.

## 5. Conclusions

The prevalence and incidence of FASD is increasing while recognition about the harmful effects of alcohol use and pregnancy often appears muted by highly strategic alcohol marketing campaigns globally. Marketing alcohol to adolescents is structured in such a way that is almost impossible to avoid exposure to frequent messages on multiple social media platforms on a daily basis at any hour of the day. Broader knowledge about the health consequences of alcohol misuse are often largely ignored in a society that cultivates personal choice, autonomy, and freedom in all aspects of life. The WHO states that over 3 million deaths per year are alcohol-related, which represents over 5% of deaths worldwide [1]. The mounting toll of disease and injury associated with alcohol use is increasing, along with the notable burden on society in relation to health care costs. The use of alcohol is seen as a personal choice and the ethical quandary of alcohol use while pregnant is not easily resolved. The problematic use of alcohol by women does not take a pause when pregnancy occurs. It is also known, however, that pregnancy can be a time when women are motivated to examine their use of alcohol. Problematic patterns of alcohol use and binge drinking in particular are known to be harmful to the developing fetus, and the use of harm reduction strategies can serve to mitigate some those effects. It is also known that women who give birth to a child with prenatal alcohol exposure are likely to have further alcohol exposed births without intervention.

What is the role and responsibility of the alcohol industry in relation to the prevention of FASD? How does one challenge alcohol use in general, and the stigma of alcohol use in pregnancy in a culture that is constantly advertising alcohol? Alcohol marketing has been determined to be directly linked and associated to drinking behavior, and alcohol is such an integral part of the social fabric of a society that to not partake comes with its own social hazards [26,29]. Young people drink to fit in with their peers and early patterns of addictive behavior often persevere and contribute to adverse health outcomes. Of course, the alcohol industry that promotes its product would simply like people to look the other way and does everything possible to glamorize the use of alcohol. A broader social determinants of health perspective must be applied when it comes to alcohol use and pregnancy, and every effort needs to be made to prevent FASD and the lifespan disabilities associated with prenatal alcohol exposure. As noted in the Yukon research in Canada, the alcohol industry has the power to target health messaging about the harmful effects of alcohol and minimize associated health problems so as not to erode profits. The erosion of health and wellbeing through alcohol use during pregnancy has implications over the life course of a child and a family. Rather than implicating mothers, who are blamed for FASD, it is time for the alcohol industry to consider the ethical implications of promoting alcohol while disregarding serious health and social costs. Alcohol use during pregnancy is directly responsible for one the most prevalent disabilities in the world. It is time for the alcohol industry to listen up.

## 6. Limitations

This work is drawn largely from the context of beverage alcohol marketing in two markets, Canada and Australia. It may be possible that other countries have differing experiences depending upon the degree of regulation that may or may not exist within the marketplace. As well, this work draws upon a specific set of regulatory and social contexts drawn from the authors’ experiences which presents specific market understandings that may also not apply in other locales.

## Figures and Tables

**Figure 1 ijerph-19-07744-f001:**
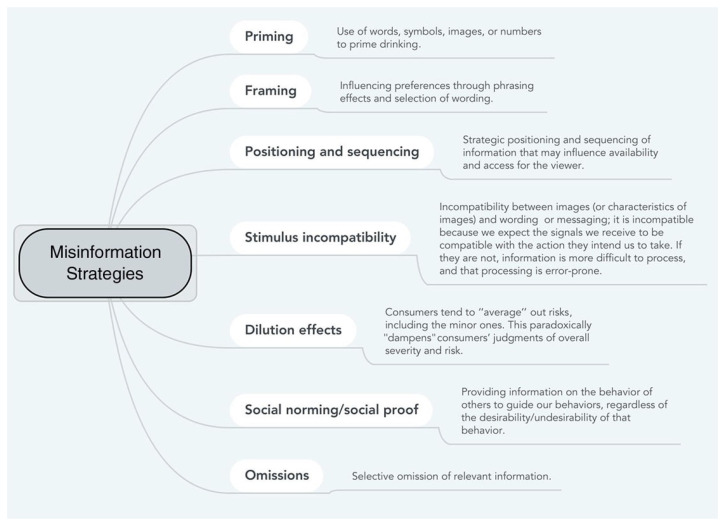
Misinformation Dark Nudges and Sludge Found in This Study (Petticrew et al., 2020) [55]. (Reprinted from Ref. [55]).

**Table 1 ijerph-19-07744-t001:** Rates of FASD per 1000 population and liters alcohol per person.

Country	FASD Prevalence/1000 Population [11]	Consumption 1/p World Population [1]
South Africa	111.1	29.9
Croatia	53.3	15
Ireland	47.5	16
Italy	45	12
Belarus	36.6	14.4
United States of America	15.2	13.7
Canada	7.9	13.8
Australia	0.6	13.4

## Data Availability

Not applicable.

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
