# Peer review of "The Alcohol Industry and Social Responsibility: Links to FASD"

_ijerph, 2022, doi:10.3390/ijerph19137744_

Round 1

Reviewer 1 Report

I would like to thank the editor of this prestigious journal for the opportunity to evaluate this study and I would also like to congratulate the researchers for the effort made and the results obtained.

First of all, I think the study is interesting and may open the door to design new strategies to better address fetal alcohol syndrome in other countries.

The material and methods section and the results section do not follow the classical structure. I think it would be interesting to create a Methods section and start by explaining the research methodology more clearly, how are the studies compared?

I also think it would be appropriate to better define the study hypotheses, the objectives pursued, in a more concrete way.

A section on results is missing, how do the two cases compare? It would also be interesting to delve more deeply into the media, to distinguish between social networks and traditional media and the difference in transmitting messages to persuade consumption in pregnancy.

The authors do not discuss the limitations of the study and many can be described, as well as what could be the significance of the study.

Overall, I think the article addresses a very important and necessary topic, but it does so in a very generic way, without evaluating or differentiating between the two campaigns or offering findings on the strengths of the campaigns.

Author Response

Please see attached chart showing changes from all reviewers

Reviewer 2 Report

This commentary article is well-presented (authors provided table and figure) and comprehensive. Several case studies are described.

Minor comment

Authors cite many articles by Popova et al. (i.e. country-specific data for Canada). However, I suggest to also implement the results from Popova et al. in Lancet Global Health 2017 regarding the global prevalence of alcohol use during pregnancy and fetal alcohol syndrome.

Author Response

Please see attached chart showing how each reviewers comments were addressed

Reviewer 3 Report

This commentary provides a good discussion around the alcohol industry and FASD. I have some minor comments on the commentary which are listed below:

1. At the beginning of the introduction, it is stated that alcohol has been widely accepted for centuries, and while true to some extent, there are cultures where alcohol is prohibited and I feel that this sentence should reflect that.

2. Line 98 discusses discrepancies around the prevalence of FASD in Australia and you would talk a little bit more here about why there might be discrepancies between the studies cited.

3. Lines 164-165 briefly mentions that there might be some cardiovascular benefits to low/moderate alcohol consumption but I think it has been well established that there are no health benefits to alcohol consumption, particularly if the comparison group are include specific types of non-drinkers.

4. Line 204 it wasn't clear which lead author you are referring to - do you mean the lead author of this commentary?

5. In aspects of the commentary, some statements could be further supported by references, such as the statement on lines 474-476.

6. Line 480 discusses the effectiveness of alcohol labels but I wonder if they could point to specific examples related to FASD and if not, you could discuss this point further.

7. It is my understanding that some of this commentary is based upon the lead author's experience of working in the alcohol industry. Line 514 discusses their experience in the industry and what their objectives were. While I do not reject those statements, I think it could be acknowledged that there might be differences across the industry in other countries etc and this is based on their own experience and might not reflect others who used to work in the industry.

Author Response

We are very  grateful for the comments and suggestions from the reviewers. Please see attached file which shows how reviewer comments were addressed

Round 2

Reviewer 1 Report

The changes made respond to my suggestions.